# 2-Year Change in Revised Hammersmith Scale Scores in a Large Cohort of Untreated Paediatric Type 2 and 3 SMA Participants

**DOI:** 10.3390/jcm12051920

**Published:** 2023-02-28

**Authors:** Georgia Stimpson, Danielle Ramsey, Amy Wolfe, Anna Mayhew, Mariacristina Scoto, Giovanni Baranello, Robert Muni Lofra, Marion Main, Evelin Milev, Giorgia Coratti, Marika Pane, Valeria Sansone, Adele D’Amico, Enrico Bertini, Sonia Messina, Claudio Bruno, Emilio Albamonte, Elena Stacy Mazzone, Jacqueline Montes, Allan M. Glanzman, Zarazuela Zolkipli-Cunningham, Amy Pasternak, Tina Duong, Sally Dunaway Young, Matthew Civitello, Chiara Marini-Bettolo, John W. Day, Basil T. Darras, Darryl C. De Vivo, Richard S. Finkel, Eugenio Mercuri, Francesco Muntoni

**Affiliations:** 1Dubowitz Neuromuscular Centre, UCL Great Ormond Street Institute of Child Health, London WC1N 1EH, UK; 2School of Health and Sports Sciences, University of Suffolk, Ipswich IP4 1QJ, UK; 3The John Walton Muscular Dystrophy Research Centre, Newcastle University and Newcastle Hospitals NHS Foundation Trust, Newcastle NE1 7RU, UK; 4National Institute for Health Research Great Ormond Street Hospital Biomedical Research Centre, London WC1N 1EH, UK; 5Pediatric Neurology Unit, Catholic University, 00135 Rome, Italy; 6Centro Clinico Nemo, U.O.C. Neuropsichiatria Infantile Fondazione Policlinico Universitario Agostino Gemelli IRCCS, 00168 Rome, Italy; 7The NEMO Center in Milan, Neurorehabilitation Unit, University of Milan, ASST Niguarda Hospital, 20162 Milan, Italy; 8Unit of Neuromuscular and Neurodegenerative Disorders, Bambino Gesù Children’s Hospital, IRCCS, 00165 Rome, Italy; 9Department of Clinical and Experimental Medicine, University of Messina, 98122 Messina, Italy; 10Center of Translational and Experimental Myology and Department of Neuroscience, Rehabilitation, Ophtalmology, Genetics, Maternal and Child Health, IRCCS Istituto Giannina Gaslini and University of Genoa, 16132 Genoa, Italy; 11Columbia University Irving Medical Center, New York, NY 10032, USA; 12Children’s Hospital of Philadelphia, Philadelphia, PA 19104, USA; 13Department of Pediatrics, University of Pennsylvania, Philadelphia, PA 19104, USA; 14Boston Children’s Hospital, Harvard Medical School, Boston, MA 02115, USA; 15Departments of Neurology TD, Stanford University, Palo Alto, CA 94305, USA; 16Nemours Children’s Hospital and University of Central Florida College of Medicine, Orlando, FL 32827, USA; 17St. Jude Children’s Research Hospital, Memphis, TN 38105, USA

**Keywords:** motor function, natural history, spinal muscular atrophy

## Abstract

The Revised Hammersmith Scale (RHS) is a 36-item ordinal scale developed using clinical expertise and sound psychometrics to investigate motor function in participants with Spinal Muscular Atrophy (SMA). In this study, we investigate median change in the RHS score up to two years in paediatric SMA 2 and 3 participants and contextualise it to the Hammersmith Functional Motor Scale–Expanded (HFMSE). These change scores were considered by SMA type, motor function, and baseline RHS score. We consider a new transitional group, spanning crawlers, standers, and walkers-with-assistance, and analyse that alongside non-sitters, sitters, and walkers. The transitional group exhibit the most definitive change score trend, with an average 1-year decline of 3 points. In the weakest patients, we are most able to detect positive change in the RHS in the under-5 age group, whereas in the stronger patients, we are most able to detect decline in the RHS in the 8–13 age group. The RHS has a reduced floor effect compared to the HFMSE, although we show that the RHS should be used in conjunction with the RULM for participants scoring less than 20 points on the RHS. The timed items in the RHS have high between-participant variability, so participants with the same RHS total can be differentiated by their timed test items.

## 1. Introduction

Spinal muscular atrophy (SMA) is an autosomal recessive neuromuscular disorder caused by mutations in the survival motor neuron 1 (SMN1) gene located on chromosome 5q leading to SMN protein deficiency [1,2,3,4]. It induces proximal muscle atrophy and weakness, leading to secondary complications including scoliosis, joint contractures, and progressive respiratory decline [4,5]. SMA is divided into types which are defined by the age of onset and the highest developmental milestone achieved. Type 1 children do not achieve the ability to sit independently, type 2 children can sit but cannot walk independently, and type 3 children achieve independent walking, but lose motor function over time and many become wheelchair dependant [4,5].

In recent years, several treatment options have been clinically proven to be effective and approved for commercial use. Both nusinersen and risdiplam are specifically designed to increase the amount of functional SMN protein by altering splicing of survival motor neuron 2 (SMN2) gene pre-mRNA. SMN2 is intact in all SMA individuals, but a single nucleotide change leads to exclusion of exon 7 from the majority of the transcript with consequent lower levels of functional SMN protein. Both nusinersen [6,7,8] and risdiplam [9,10] have been studied in symptomatic type 1 and pre-symptomatic cohorts, as well as in type 2 and 3 SMA, and significant benefits have been demonstrated, with transformative changes especially when administered close to disease onset or pre-symptomatically [3,6,7,8,10,11]. However, functional improvement or stabilisation of function in more advanced and chronic stages of the disease require more careful documentation, and comparison with the natural history in SMA types 2 and 3 is required. 

The Hammersmith Functional Motor Scale Expanded (HFMSE) is a clinical outcome assessment designed and validated to assess gross motor function in SMA [12]. The Revised Hammersmith Scale (RHS) was developed to address discontinuity in the HFMSE [13], and several items were adapted and added from the North Star Ambulatory Assessment (NSAA) [14] and the Children’s Hospital of Philadelphia Infant Test of Neuromuscular Disorders (CHOP-INTEND) [15] to increase the sensitivity of the scale in the strongest and weakest patients, respectively. From the NSAA, a scale validated and widely used in Duchenne muscular dystrophy, items relating to one legged standing, hopping, and climbing/descending box steps were included alongside the two timed items (the rise from floor (RFF) and 10 metre walk/run test (10MWR)). From the CHOP-INTEND, the “Adduction from Crook Lying” item was included. The Revised Upper Limb Module (RULM) was specifically designed to capture upper limb function in SMA and is used as an outcome in ongoing clinical trials. 

Due to the availability of multiple outcome measures to assess patients with SMA types 2 and 3, their comparative strengths and weaknesses need to be understood. Recent therapeutic innovation and the increased availability of disease-modifying drugs have led to a change in phenotypes, with the majority of children with SMA now on a treatment. The cohort analysed here is one of the largest natural history cohorts of SMA types 2 and 3 available. The increasing availability of treatments makes it crucial to understand the sensitivity to change of available outcome measures, to aid trial design and inform clinical care. The data presented will provide reference data to detect changes in treated SMA 2 and 3 patients. 

### Aims

We aim to characterise the change in RHS scores over a two-year period by age, motor function, and total RHS score in a large international cohort of untreated SMA 2 and 3 participants. We aim to contextualise these change scores by providing the corresponding change in the HFMSE score. The aim of this longitudinal, multicentre natural history study is to demonstrate how the RHS score can be used in conjunction with other functional measures such as the RULM and the RHS timed items to enhance the understanding of this cohort’s disease progression and detect changes with treatments.

## 2. Materials and Methods

### 2.1. Inclusion Criteria

The participants included in this analysis are recruited from the International SMA Consortium (iSMAC) natural history studies (SMA REACH UK, PNCRN USA and Italian Telethon) [16]. All participants had a genetically confirmed diagnosis of SMA type 2 or 3, were receiving SMA Standards of Care treatment [17,18,19], had no previous involvement in clinical trials, and had at least two RHS assessments performed between the 17 March 2015 and the 29 July 2019.

### 2.2. Scales

RHS, HFMSE, and RULM assessments were conducted by experienced neuromuscular physiotherapists who were part of, or trained by, iSMAC. RHS, HFMSE, and RULM scores were collected in clinics approximately every 6 months, as recommended in the standard of care [18,19]. The RHS is a 36-item ordinal scale with a maximum score of 69 points (33 items are scored 0–2, and three 0–1). The HFMSE is 33-item ordinal scale with a maximum score of 66 points. The RHS and HFMSE can be scored simultaneously, as many items are similar or a perfect match between scales. The RULM is a 20-item ordinal level scale (including a separately scored entry item) which captures proximal, mid-level, and distal arm performance with a maximum score of 37 points.

### 2.3. Analysis

Participants without a known SMA type, gender, RHS total score, and HFMSE total score were excluded from the analysis. As the data were collected longitudinally in clinic, the participant visits were not scheduled uniformly at six-month intervals. Therefore, visits that were completed ±3 months were accepted for analysis. Additionally, to maximise participant populations, every participant assessment could act as a baseline [20]. 

RHS medians and interquartile ranges are presented. Significance testing for the RHS and HFMSE change scores was completed using a sign test for the median = 0, with a significance level of 5%. For some groups, it was not possible to compute the sign test due to low sample size/low number of non-zero change scores. Means and standard deviation (SD) values for the change scores are presented in the Appendix A, and, here, the *p*-values are calculated using a *t*-test. Participants were stratified according to SMA type, defined by peak motor function attainment (for SMA type 2 vs. SMA type 3) and symptom onset (for SMA type 3a vs. SMA type 3b). The World Health Organisation (WHO)-derived functional groups were determined based on previous published work [21], which grouped participants based on their WHO motor function, with scores of 2 (crawling), 3 (standing with assistance), 4 (standing independently), or 5 (walking with assistance) coded as the “transitional group”. Of note here, this scale is not ordinal, and patients who, for example, could not crawl but could stand with assistance were classified at a 3 instead of a 1. Change scores were stratified by type and age as follows: < 5, 5–7, 8–13, and 14–18 years, in order to align with previous research on the HFMSE which used similar age groups [22,23]. Additionally, the change scores were stratified by baseline motor function, which were described using quintiles of the RHS total score across the whole population. The quintiles of the RHS total score were calculated using all the data and were defined as follows: Quintile 1 (Q1)-scores from 0–4, Quintile 2 (Q2)-scores from 5–9, Quintile 3 (Q3)-scores from 10–18, Quintile 4 (Q4)-scores from 19–42, Quintile 5 (Q5)-scores from 43–69. 

To model the relationship between the RULM and the RHS, and the RHS timed items and the RHS total, a random effects model was used to adjust for the between-participant correlation. The timed tests were only considered as valid if the participants scored >0 on the corresponding item. For the timed tests, a linear model was considered with a person-specific intercept only. To define the RHS total score, which was most predictive of the performance of the RHS timed items, the receiver operating characteristic (ROC) was used, which trades off the sensitivity and specificity rates of potential cut offs. The ROC curve is not shown here. When jointly considering the RULM and the RHS and the timed tests and the RHS, a piecewise linear model with one knot was used. This creates two joined straight lines to represent an inflection point in the relationship. The position of the inflection point was identified using a grid-search (fitting the model with each possible breakpoint from 1–67), and the value that minimised the Akaike information criterion (AIC) (which is a trade-off between model complexity and goodness of fit) was chosen. When considering the timed items, the linear model was found to minimise the AIC compared to a piecewise linear model with one knot. All analysis was completed in R (version number 3.6.0).

## 3. Results

### 3.1. Participants

This analysis consisted of 177 participants assessed at 586 time points (an average of 3.3 assessments per participant). Participants were recruited from seven sites globally and the populations at each site were varied.

The majority of participants included in this analysis were SMA type 2 (62%), with 33% SMA type 3a and 5% type 3b. The full range of the RHS was observed in this cohort, with a median score of 12. An overview of the medians, interquartile ranges, and ranges observed in this population are available in Table 1. Female patients made up 47% of the cohort, with significantly stronger RHS scores observed in the females compared to the males (*p* = 0.007). In the 149 patients where spinal surgery status was known, 23 (15%) had undergone spinal surgery. In these patients, the median RHS score was lower and the median age was higher.

### 3.2. 2-Year Change in RHS and HFMSE

The full RHS and HFMSE change scores are presented in Table 2. We observe relative stability in the SMA type 2 and 3b groups. However, in the SMA type 3a group, there is a trend towards a slight decrease in scores, which is only significant at 18 months (−2 in the RHS (*p* = 0.027), −1.5 in the HFMSE (*p* = 0.01)). When considering the change score by age group, the under-5-year age group are the only participant subgroup who display a positive significant change score across all time points of both the RHS and HFMSE. In the 5–7 years subgroup, there is, on average, a trend of decline from 18 months, although this is not significant. In the 8–13 years subgroups, we see a decrease in RHS scores at all time points, which are significant at 6, 12, 18, and 24 months (*p* = 0.001, *p* < 0.001, *p* < 0.001, and *p* = 0.005, respectively). The 14–18 years subgroup display a mild, non-significant trend towards decline at 6 and 12 months. Grouping change scores by motor function, the strongest difference is observed in the transitional group, where the average change is significantly negative at 12, 18, and 24 months (*p* < 0.001, *p* < 0.001, and *p* = 0.016 for both the RHS and HFMSE). It is worth noting that this is the smallest subgroup.

When considering the change in RHS and HFMSE with respect to cross-tabulated SMA types and age (full results shown in Table 3), we found that in the under-5 age group, the SMA 2 subgroup showed increasing RHS and HFMSE scores, which were significant at all follow ups (*p* = 0.01, *p* < 0.001, *p* = 0.002, and *p* = 0.003 for the RHS; *p* = 0.06, *p* = 0.002, *p* = 0.015, and *p* = 0.001 for the HFMSE). Notably, at 24 months, there was a median 1-point increase in the RHS, with 75% of assessments for which there was a 24-month follow up having at least a 1-point increase in the RHS. In the under-5 age group, the SMA 3a group also showed a trend towards increasing score at and after 12 months, although this was only significant at 6 months in both the RHS and HFMSE (median 6 (*p* = 0.03) and 3 (*p* = 0.06), respectively). 

In the 5–7 age group, there is a decline in the RHS scores for the SMA 2 subgroup that is significant at 2 years (−2 points, *p* = 0.021), but this was not significant in the HFMSE. There is no significant change in the 3a participants in this age group, although there is a trend towards decline in the RHS and not the HFMSE.

In the 8–13 age group, there is a clear decline in the SMA 2 participants in the RHS and the HFMSE, with a significant median decline of -1 in both the RHS and the HFMSE at 12 months (*p* = 0.003 and *p* < 0.001, respectively). In this group, the median 24-month change in the RHS is −1.5 (*p* = 0.019), and in the HFMSE is −3 (*p* = 0.004). The same changes were also seen in the SMA 3a participants between the age of 8 and 13 years at baseline, with a significant decline in both the RHS at 6, 12, and 18 months (*p* = 0.003, *p* = 0.004, and *p* = 0.013, respectively) and in the HFMSE at 6, 12, 18, and 24 months (*p* = 0.007, *p* = 0.001, 0.001, and *p* = 0.035, respectively). At 24 months, the median RHS change is -9, and the median HFMSE change is -6. In the over-14s, there is an overall stability in the SMA 2s and 3bs, and a trend towards decline in the 3as, although the significance could not be assessed due to the small numbers. 

When considering the change scores by both age group and motor function at baseline (as in Table 4), we observed increasing RHS and HFMSE scores in the under-5 sitters, which is significant at 6, 12, 18, and 24 months (*p* = 0.014, <0.001, <0.001, and <0.001 for the RHS, respectively; and 0.018, 0.002, 0.004, and <0.001 for the HFMSE, respectively). There is an average increase of 1.5 and 2.5 points in the RHS and HFMSE, respectively, at the 24-month time points. This contrasts with the sitters in the 5–7, 8–13, and 14–18 age groups, who display negative change scores across all time points in both the HFMSE and RHS. In the sitters aged between 8 and 13 at baseline, there is a significant decline at 1-year of -1 in the RHS (*p* = 0.002) and -2 in the HFMSE (*p* < 0.001), respectively. In the transitional group, there is a trend of decline in all age groups, with notable changes in the 5–7 and 8–13 age groups, although significance could not be assessed due to small sample sizes. In the walkers, there is a clear improvement in the under-5s, which is significant at 6 months in both the RHS (5, *p* = 0.006) and the HFMSE (3, *p* = 0.012). 

The full change scores of the cross tabulated data by age group and RHS total score are presented in Table 5. In those scoring in the lowest quartile of the RHS (RHS score 0–4) before the age of 5, there is an increase in both the RHS and the HFMSE scores. The change in the RHS is significant at 6, 12, and 18 months (*p* = 0.002, 0.001, and 0.0031, respectively), but due to small sample size, it was not possible to calculate the significance at 24 months. Of relevance, this group do not show significant changes in the HFMSE. In other age groups, participants scoring in the lowest quartile had no detectable change (the absolute average change was ≤1) at any follow-up on either the RHS or the HFMSE. In the Q2 group (RHS score 5–9 at baseline), there was an increase in both the RHS and the HFMSE scores in the under-5 group, which was significant at 6, 12, and 24 months in the HFMSE (*p* = 0.043, 0.004, and 0.002, respectively), and at 12 and 18 months in the RHS (*p* = 0.004 and 0.004, respectively). It was not possible to assess the significance of the change at 24 months due to small sample sizes. In this group, the median 12 months change was 2 points on both the RHS and the HFMSE, whereas at 18 months, it was 5.5 and 6, respectively. In the Q2 group, between the ages of 8 and 13, there was a trend towards decline in both the RHS and HFMSE. In the Q3 participants (RHS score 10–18 at baseline), there was a decline in the 8–13 age group, which was significant at 12 and 18 months in both the HFMSE (*p* = 0.007 and 0.012) and the RHS (*p* = 0.035 and *p* = 0.021). In the Q4 group (RHS score 19–42 at baseline), between the ages of 8–13, there was a significant decline in the RHS and HFMSE at 6, 12, and 18 months (for the RHS, *p* = 0.004, 0.001, and 0.001, respectively; for the HFMSE, *p* = 0.019, *p* < 0.001, and 0.002, respectively). In this cohort, the median 2-year change is -9 on the RHS and -6 on the HFMSE. In the Q5 group (RHS score 43–69 at baseline), there was a trend towards decline in the RHS and HFMSE, but it was not significant.

### 3.3. Ceiling and Floor Effect

Five participants, across 11 assessments, scored a 0 on the RHS, all SMA type 2 (median age 13.8 years (IQR: 12.2, 16.5)). Spinal surgery status was only known for eight of the assessments; of these, six occurred after spinal fusion surgery (75%). Eleven participants, all SMA type 2, across 19 assessments, scored a 0 on the HFMSE (median age 13.1 years (IQR: 10.6, 15.5)). Twelve of these assessments occurred after spinal fusion surgery (63% of the assessments where spinal fusion surgery was known), with zero assessments having missing spinal fusion surgery data. 

On the occasions where participants scored a 0 on the RHS, the majority also scored a 0 on the HFMSE (75%). However, there was a lower rate of floor assessments in the RHS compared to the HFMSE, with participants in two thirds (13/19) of the assessments in which they scored a 0 on the HFMSE having a non-zero score on the RHS. The details of this are presented in Table 6. Here, the seven participants who scored a 1 all achieved a 1 on the Item 4—Adduction from Crook Lying item. For the six participants who scored a 2 on the RHS whilst scoring a 0 on the HFMSE, they all scored a 2 on Item 4—Adduction from Crook Lying.

Only two participants scored a maximum score of 69 in the RHS, a 6.3-year-old SMA 3a participant and a 15.2-year-old SMA 3b participant. Both only achieved this during one assessment, and both also scored the maximum of 66 on the HFMSE. These were the only participants who scored the maximum on the HFMSE.

### 3.4. Timed Tests

The timed portion of Item 19, runs 10 m, was recorded for 95 assessments; however, the corresponding item score was 0 for 7 (7%) of the assessments, and so these were removed. This yielded 88 assessments for 31 patients. The timed portion of Item 25, rise from floor, was recorded for 100 assessments; however, the corresponding item score was 0 for 19 (19%) of the assessments, and so these were removed. This yielded 81 assessments for 28 patients. 

The average time for Item 19 (runs 10 m) when achieving a RHS total score of 50 was 7.91 s, and for every increase of one-point in RHS total score, this time reduced by 0.17 s. The average time for item 25 (rise from floor) when achieving a RHS total score of 50 was 10.18 s, and for every increase of one-point in RHS total score, this time reduced by 0.36 s. When comparing trends of RHS total score with the two timed test items, we see that for participants achieving above a 60 on the RHS total, there is about 5 s of variation in the time for run and nearly 20 s in variation in the rise from floor time. The rise from floor time is noticeably more variable than the time for run, as can be seen in Figure 1. This suggests that higher granularity can be achieved by looking at the rise from floor. Additionally, the rise from floor shows more of a person-specific effect, where some participants have higher rise-from-floor times across all time-points. This may suggest that the rise from floor captures aspects of strength and function that are at least partially independent of the RHS total score. A possible contributor to these differences might also be related to the lower limb contractures, muscle imbalances, and the different distribution of muscle weakness that can impact strategy selection and be observed between participants apparently performing at a similar level when assessed with a functional scale.

A score of 42 or above on the RHS was the most predictive (in terms of trading-off specificity and sensitivity) of the ability to complete Item 19, run 10 m. Of the 463 assessments where participants scored less than 42, participants in 1% of assessments (six) completed the timed test. In 70% of assessments where participants scored a 42 or more, the timed test was completed (83 timed test completed, 35 not completed). 

A score of 44 or above on the RHS was the most predictive (in terms of trading-off specificity and sensitivity) of the ability to complete Item 25, rise from floor. Of the 477 assessments where participants scored less than 44, participants in <1% of assessments (three) completed the timed test. In 72% of assessments where participants scored a 44 or more, the timed test was completed (78 timed test completed, 31 not completed).

### 3.5. RULM and RHS

The RULM score was recorded for 86 participants at 226 time points. There is a clear trend linking the RULM and the RHS, with a score of 0 on the RHS equivalent to an average score of 10.33 on the RULM. The RULM score then increases linearly with the RHS, with a one-point increase on the RHS equivalent to 0.74 points on the RULM, up to a total score of 22 on the RHS. After this, the slope is shallower, as the assessment scores are impacted by the ceiling effect of the RULM. Here, a one-point increase on the RHS corresponds to a 0.21-point increase on the RULM. This trend is demonstrated in Figure 2. Our data suggest that the RULM provides more information than the RHS in assessments where the participants have relatively low scores in the RHS: for scores under 20 on the RHS, there is a range of 37 points on the RULM. Similarly, in those scoring under 10 on the RHS, there is a range of 25 points on the RULM, and in those scoring 0 on the RHS, there is an 18-point range on the RULM. 

## 4. Discussion

The RHS and HFMSE are two scales designed to capture motor function in SMA. Whereas the HFMSE captures the physical abilities of SMA type 2 and type 3 participants with limited/no ambulation, the RHS was developed to extend the range of functional abilities captured by the HFMSE at both the lower (including the Adduction from Crook Lying item adapted from the CHOP-INTEND) and the upper end (including the Stand on one Leg, Hop, Climb, and Descend Box Step, and the timed items adapted from the NSAA). Consequently, the RHS was designed to assess physical abilities of very weak SMA type 2 participants who are no longer able to sit through to very strong, ambulant participants with SMA type 3. However, longitudinal comparative analysis of these two scales is limited, and correlation of the RHS with other functional scales capable of capturing aspects of function related to the upper limb (RULM) have not been reported so far.

In the non-sitters, our findings of no-change at 12 and 24 months is in line with what has been previously reported with the HFMSE [20]. This remains a limitation for both scales, which can be addressed by the inclusion of additional assessments of upper limb function. Indeed, we found that the RHS is highly correlated with the RULM, similarly to the HFMSE [24]. Additionally, our results suggest that the RHS can be enhanced by the RULM when considering the variation in the weaker participants who are scoring under 20 on the RHS. In the non-sitters, who typically achieve between 1 and 3 on the RHS, the RULM total scores ranged from 0–20, providing much more sensitivity for disease progression. We suggest that the RHS is performed alongside the RULM in these cases rather than using a gross motor scale such as the CHOP-Intend [15]. As the CHOP-Intend was developed for use in infants, items 15 and 16, which are performed in ventral suspension, are not appropriate for older patients, which leads to incomplete assessments.

Our findings show that any improvement of scores on the RHS or the HFMSE over time would represent a positive divergence from the natural history and could allow to assess therapeutic response even in this patient population. It is worthy to note that the floor effect observed for the RHS occurs only in this subpopulation of non-sitters, and that the RHS exhibits less floor effect than the HFMSE, with two thirds of the participants who achieved 0 on the HFMSE achieving a 1 or a 2 on the Adduction from Crook Lying item on the RHS. Our data suggest, therefore, that by considering the RHS, individuals who otherwise are indistinguishable on the HFMSE can be split into three groups based on their Adduction from Crook Lying item score. 

In this study, we split the population that has been defined as sitters in previous studies with the HFMSE into two separate groups: one group included those patients whose maximum motor function was sitting (sitters); and another that included those patients who were also able to crawl, stand with support, walk with support, or stand independently (transitional). Previous work from our group has shown that the transitional group have RHS scores significantly different from the sitters group [21]. In a previous study on the HFMSE, where the group of sitters was considered to include both what we have defined as sitters and transitional in the present study, Coratti et al., 2020 [20] found that the 12- and 24-month HFMSE scores had a change in mean of −0.83 and −1.99, respectively. In our study, when we considered the sitters broken down by age groups, we found that there was significant change in median scores of 1.5 at 24 months for the under-5 age group, whereas there was a non-significant trend towards decline of -1 and -2 at 24 months in the 5–7 and 8–13 age groups, respectively. These findings suggest that an average gain in RHS or HFMSE score over time in sitters over the age of 5 would signify treatment response in a treated patient population. 

When considered in isolation, the newly defined transitional group was the group who displayed greatest change over time. It is worth noting that a smaller proportion of our participants are in this group, but despite the heterogeneity of the group in terms of age, these participants display relatively homogeneous decline. We observed a trend towards decline in this group in all age groups beyond the age of 5, but none of the sample sizes were large enough to complete significance testing. This transitional group is likely a population of interest for therapeutic research, as they would be most likely to display a detectable treatment effect in the shortest timeframe.

Patients who scored between 10 and 18 on the RHS at baseline (Q3), represent the mid-to-strong sitters and the weaker transitional patients. In this group, we found increasing RHS scores in the under-5 age group, and a decline in both the RHS and HFMSE in the 8–13 age group that was significant at 12 and 18 months in the RHS and the HFMSE. In patients scoring between 19 and 42 on the RHS at baseline (Q4), which represent the strongest sitters, the transitional patients, and the weaker ambulant patients, we found improving scores in the under-5 age group, and a significant decline in the 8–13 age group in both the RHS and HFMSE, with a median 2-year change of −9 on the RHS and of −6 on the HFMSE. In the SMA 3b group, we observed stability at 6, 12, 18, and 24 months, although the number of SMA 3b participants in this study was too small to consider the change with respect to age group. 

A ceiling effect was identified in <1% of RHS and HFMSE assessments (and <1% of participants) in the present study [25]. Both participants in this analysis scoring the maximum on the HFMSE also scored maximum in the RHS. It was not possible, therefore, to compare the relative strength of the score in the very strongest SMA patients, as none were included in this sample. However, the RHS includes two timed tests which were designed to increase the sensitivity of the scale for the strongest patients. The relationship between the RHS and the two timed tests was linear on average, and we found that the timed tests allowed to discriminate between patients having similar RHS total scores. This was particularly true for the rise from floor time, where the between-person variability was very high and the within-person variability was lower. It is likely that this between-person variability reflects the differences in pattern of muscle involvement in individual participants affected by SMA irrespective of the subtype. Our findings, therefore, suggest that it is possible to differentiate patients who are scoring similarly on the RHS by using the rise from floor time, thus increasing the sensitivity of the scale in patients achieving a total score of over 40. Additionally, patients scoring a RHS total score above 41 were most likely to perform the walk 10 m item, and this occurred in 70% of assessments. Similarly, the rise from floor item was most likely to be performed in assessments where the participant had an RHS total score of at least 44. It is worth noting that in 7 and 19% of assessments where the walk 10 m and rise from floor items were reported respectively, the corresponding items score was 0, and these times were discarded. This high rate is potentially due to the newness of the RHS scale, although it should be noted that these times could still be informative, particularly the rise from floor, where a 0 corresponds to rise with furniture. In the SMA 3a and 3b participants, it would be important to understand the change in timed test by age group, but this was not possible with this data due to small sample sizes. 

One limitation of this study is that by splitting the participants into age groups, and not considering instead age as a continuous variable, it is possible that the groups chosen do not fully represent the participants. The groups here are nevertheless similar to groups used in papers analysing the HFMSE and reflect the peak in motor function observed at 5 years in the SMA 2s, and at 7 years in the SMA 3s [24]. In order to allow analysis of the change scores by motor function category and baseline RHS, the groups that have previously been used (i.e., <5, 5–13, >13 for SMA 2s age [20,23,26,27]; and <5, 5–7, 8–14, >14 for SMA 3s [23,28]) were merged to create the following groups: <5, 5–7, 7–13, 14–18. In this analysis, only children were included, but it will be important to separately analyse the natural history of the RHS in adult SMA in the future [23]. Finally, as this population was selected from the full collected cohort based on completeness of the full RHS, with non-missing items, we cannot exclude that the participants included in this analysis will be biased towards a stronger population, as it might be that some participants having missing items because they were not able to complete the items were excluded from the analysis. The inclusion of missingness codes for each item in future prospective data collections, with options such as “not completed due to injury” vs. “not completed due to lack of time”, could address this limitation.

Despite these limitations, this study represents the most comprehensive natural history dataset to contextualise disease progression in different subgroups of the SMA 2 and SMA 3 populations, and natural history data are becoming less available due to increasing therapeutic availability worldwide. 

## 5. Conclusions

This study describes change over time in RHS scores across SMA type, age, ambulatory status, and newly described WHO-derived functional types. We showed that the RHS is more effective at differentiating non-sitter participants and has a reduced floor effect when compared to the HFMSE. In addition, our findings confirm that the RULM should be used in conjunction with the RHS for the weakest patients. In particular, RULM score should be used to differentiate participants scoring less than 20 on the RHS. We also showed that change in RHS over time is as sensitive as the HFMSE, and that they can both detect change in certain type, age, and motor function subgroups. However, the timed tests included in the RHS mean that it is more able to discriminate between patients scoring similarly on the RHS total, a benefit over the HFMSE. This information is an important factor in improving clinical trial designs, informing future patient clinical guidelines, and assisting in the interpretation of results of medical interventions in SMA types 2 and 3.

## Figures and Tables

**Figure 1 jcm-12-01920-f001:**
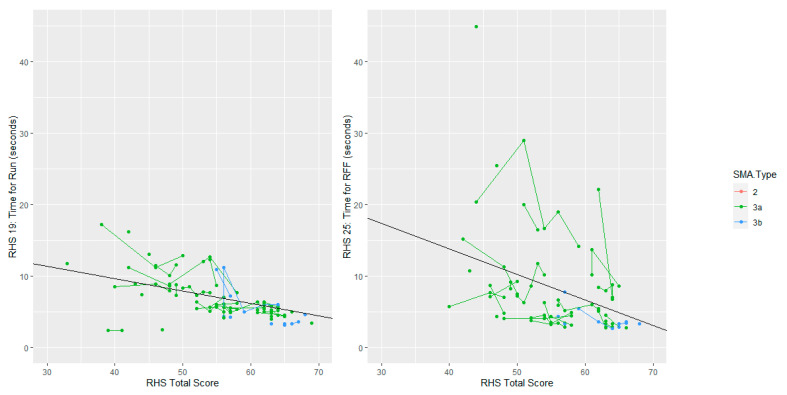
Trend in RHS Timed tests by RHS total score, with overlayed mean line. Lines connect the same patients over multiple time points.

**Figure 2 jcm-12-01920-f002:**
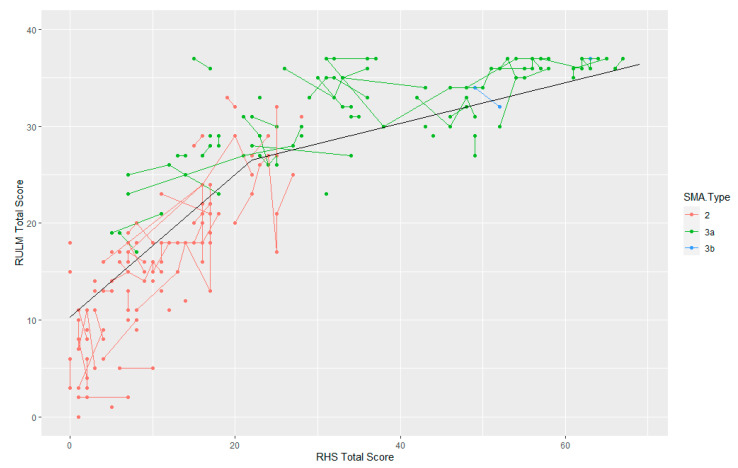
Trend in RULM compared to the RHS with overlayed piecewise linear mean line.

**Table 1 jcm-12-01920-t001:** Patient first visit RHS and HFMSE scores, and ages by SMA type, age, and motor function level. *N*: number of participants, *M*: number of assessments, *IQR*: inter-quartile range.

Grouping		N	RHS Median (IQR)	RHS Range	HFMSE Median (IQR)	HFMSE Range	Age Median (IQR)	Age Range
All		177	12 (6–34)	0–69	18 (7–40)	0–66	7.7 (4–10.6)	1–17.5
SMA Type	2	110	7 (4–11)	0–27	9 (4–18)	0–39	6 (3–9.6)	1–17.5
	3a	58	40 (25–48)	7–67	47 (35–51)	7–63	8.8 (5.3–11.5)	2.7–15.7
	3b	9	59 (49–64)	31–69	60 (53–63)	36–66	13.3 (9.4–15.2)	4.2–17.1
Age Group	<5	60	10 (6–23)	3–52	16 (9–34)	3–53	3.2 (2.2–4)	1–4.9
	5–7	33	15 (7–46)	3–67	22 (10–49)	3–63	6.3 (5.5–7.3)	5–7.9
	8–13	64	16 (5–38)	0–65	21 (6–44)	0–63	9.8 (9.1–11.8)	8–13.9
	14–18	20	6 (3–27)	0–69	7 (3–38)	0–66	14.9 (14.6–16.1)	14.2–17.5
WHO-derived Functional Group	Non–Sitter	22	2 (1–4)	0–8	2 (0–4)	0–12	10.3 (6.4–12.6)	1–16.5
	Sitter	98	8 (6–15)	2–26	12 (7–20)	2–40	5.8 (3.3–9.6)	1.2–17.5
	Transitional Group	11	26 (25–29)	18–59	36 (34–38)	24–61	8.7 (6.3–13.1)	2.8–17.1
	Walker	46	46 (41–58)	27–69	49 (47–58)	35–66	8.1 (4.8–10.1)	2.7–15.2
Ambulation	Non– Ambulant	131	8 (4–15)	0–59	11 (5–21)	0–61	7.6 (3.8–10.9)	1–17.5
Gender	Male	94	10 (5–24)	0–69	14 (7–34)	0–66	7.9 (4.5–11.1)	1–17.5
	Female	83	15 (6–40)	0–67	21 (9–45)	0–63	7.3 (3.6–9.9)	1.2–16.6
Spinal Surgery	Yes	23	5 (2–10)	0–59	6 (2–13)	0–61	13.5 (10.2–14.9)	2.2–17.5
	No	126	14 (6–36)	0–69	20 (9–41)	0–66	6.3 (3.6–9.5)	1–15.7
RHS Total Score Group	Q1	36	3 (2–4)	0–4	3 (1–4)	0–8	9.8 (4.9–13.8)	1.5–16.6
	Q2	38	6 (6–7)	5–9	9 (7–11)	4–16	6.1 (2.5–9.6)	1–17.5
	Q3	38	14 (10–16)	10–18	19 (15–23)	11–29	5.1 (3.7–8.6)	1.5–15.7
	Q4	34	28 (25–39)	20–42	38 (34–46)	27–49	9.3 (4.6–10.9)	2.2–16.3
	Q5	31	56 (46–64)	43–69	56 (50–62)	36–66	7.9 (5.5–10.1)	3.3–17.1

**Table 2 jcm-12-01920-t002:** Up to 2-year median change in the RHS and HFMSE broken down by SMA type, age, and motor function group. *N*: number of participants, *M*: number of assessments, *IQR*: inter-quartile range.

Follow up			6-Month				12-Month				18-Month				24-Month			
			M	N	RHS	HFMSE	M	N	RHS	HFMSE	M	N	RHS	HFMSE	M	N	RHS	HFMSE
ALL		Median (IQR)	316	124	0 (−1, 1)	0 (−2, 1)	244	112	0 (−2, 2)	0 (−3, 2)	137	82	0 (−4, 2)	−1 (−5, 1)	101	63	0 (−4, 2)	0 (−5, 3)
		*p*-value			0.365	0.603			0.834	0.211			0.223	0.015			0.53	0.598
SMA Type	2	Median (IQR)	188	77	0 (−1, 1)	0 (−1, 1)	144	63	0 (−1, 1)	0 (−2, 1)	75	43	0 (−2, 1.5)	0 (−4, 1)	58	33	0 (−2.75, 1)	0 (−4.75, 2)
		*p*-value			0.419	0.93			1	0.188			1	0.117			0.568	0.672
	3a	Median (IQR)	102	39	−1 (−4, 2)	−0.5 (−3, 2)	85	44	−1 (−4, 2)	−1 (−4, 2)	52	34	−2 (−6, 1.25)	−1.5 (−7, 1.25)	33	27	−1 (−9, 4)	−1 (−8, 3)
		*p*-value			0.079	0.203			0.368	0.434			0.036	0.04			0.728	0.86
	3b	Median (IQR)	26	8	−1 (−2.75, 0.75)	0 (−1, 1)	15	5	2 (−2.5, 4)	1 (−1.5, 2.5)	10	5	2 (−1.75, 4.5)	0.5 (−1, 2.75)	10	3	0.5 (−1.75, 3.75)	0 (−1, 4.5)
		*p*-value			0.189	0.824			0.424	0.424			0.754	1			1	1
Age	<5	Median (IQR)	100	44	0 (−0.25, 2)	0 (−1, 2)	80	37	1 (0, 4)	1 (0, 4)	39	28	3 (0, 5)	3 (0, 6)	29	17	2 (1, 6)	3 (1, 8)
		*p*-value			0.022	0.014			<0.001	<0.001			0.001	0.006			<0.001	0.001
	5–7	Median (IQR)	66	33	0 (−1, 1)	0 (−1, 2)	48	30	0 (−2, 2)	0 (−3, 2)	24	16	−1.5 (−4, 0.25)	−1 (−4, 1)	22	17	−1.5 (−4, 1.5)	−2 (−4, 2)
		*p*-value			1	0.488			0.875	1			0.115	0.263			0.078	0.664
	8–13	Median (IQR)	118	53	−1 (−2, 1)	−1 (−2.75, 0.75)	97	47	−1 (−4, 0)	−2 (−5, 0)	63	34	−2 (−5.5, 0)	−4 (−7, 0)	43	30	−2 (−8, 0)	−4 (−9, 0)
		*p*-value			0.01	0.001			<0.001	<0.001			<0.001	<0.001			0.005	0.001
	14–18	Median (IQR)	32	14	−1 (−1.25, 1)	0 (−2, 0)	19	14	−1 (−2.5, 0.5)	0 (−2, 0.5)	11	8	0 (−0.5, 0.5)	0 (−1, 1)	7	2	0 (−0.5, 2)	0 (−1, 3)
		*p*-value			0.185	0.359			0.302	0.581			1	0.727				1
WHO-derived Functional Group	Non-Sitter	Median (IQR)	14	11	0 (−0.75, 0.75)	0 (0, 1)	15	12	0 (−1, 1.5)	0 (−1, 0.5)	9	7	0 (−1, 0)	0 (0, 0)	4	4	−0.5 (−1, 0.25)	0 (−0.25, 0.25)
		*p*-value			1	0.219			1	0.754								
	Sitter	Median (IQR)	188	77	0 (−1, 1)	0 (−2, 1)	138	63	0 (−1, 1)	0 (−2, 1.75)	68	40	0 (−3, 2)	−1 (−4, 1)	53	30	0 (−2, 2)	0 (−5, 3)
		*p*-value			1	0.261			0.702	0.259			0.892	0.111			1	1
	Transitional	Median (IQR)	24	11	−1 (−3.25, 1)	−1 (−4, 1)	21	13	−3 (−5, −2)	−3 (−6, −1)	14	9	−6 (−7, −2.5)	−6 (−9, −3)	7	5	−7 (−7, −4.5)	−4 (−7.5, −4)
		*p*-value			0.383	0.383			<0.001	0.001			<0.001	<0.001			0.016	0.016
	Walker	Median (IQR)	90	33	−0.5 (−3, 2)	0 (−2, 1.75)	70	34	1 (−3.75, 4)	1 (−3, 3)	46	30	0 (−3.75, 4)	0 (−4, 3)	37	27	0 (−9, 4)	0 (−5, 3)
		*p*-value			0.26	0.567			0.321	0.215			1	0.878			1	1
Ambulation	Non-Ambulant	Median (IQR)	226	95	0 (−1, 1)	0 (−2, 1)	174	82	0 (−2, 1)	0 (−3, 1)	91	55	0 (−3.5, 1)	−1 (−5, 1)	64	38	0 (−3, 1)	0 (−5, 2)
		*p*-value			0.813	0.27			0.312	0.015			0.125	0.005			0.419	0.504

**Table 3 jcm-12-01920-t003:** Up to 2-year median change in the RHS and HFMSE cross-tabulated by SMA type and age. *N*: number of participants, *M*: number of assessments, *IQR*: inter-quartile range.

Age		<5				5–7				8–13				14–18			
SMA Type		6 m	12 m	18 m	24 m	6 m	12 m	18 m	24 m	6 m	12 m	18 m	2 4 m	6 m	12 m	18 m	24 m
2	M	89	67	29	22	39	24	12	11	51	46	30	24	9	7	4	1
	N	37	28	19	12	20	15	8	7	27	23	15	16	5	6	3	1
	RHS Median (IQR)	0 (0, 2)	1 (0, 2)	2 (0, 5)	1 (1, 4.5)	0 (−1, 0)	−0.5 (−2, 0.25)	−2 (−4, 0)	−2 (−4, −1)	0 (−1, 1)	−1 (−2, 0)	−1 (−3.75, 0)	−1.5 (−4.25, 0)	0 (−1, 1)	0 (−1, 0)	0 (0, 0)	
	*p*-value	0.01	<0.001	0.002	0.003	0.405	0.238	0.18	0.021	0.487	0.003	0.023	0.019				
	HFMSE Median (IQR)	0 (0, 2)	1 (0, 4)	2 (0, 5)	2 (1, 5.75)	0 (−2, 1)	−1 (−3.25, 0.25)	−2 (−5, 0.25)	−4 (−4.5, −2)	0 (−2, 0)	−1 (−3.75, 0)	−4 (−6, 0)	−3 (−8, 0)	0 (0, 0)	0 (−0.5, 0)	−0.5 (−1, 0)	
	*p*-value	0.006	0.002	0.015	0.001	0.442	0.052	0.344	0.065	0.024	<0.001	<0.001	0.004				
3a	M	10	13	10	7	26	22	12	11	58	44	28	15	8	6	2	0
	N	6	9	9	5	12	14	8	10	23	21	16	12	4	5	2	0
	RHS Median (IQR)	−0.5 (−3.25, 2.25)	6 (4, 8)	4 (−1.5, 5.75)	6 (4.5, 9)	0.5 (−3.5, 3.75)	0.5 (−2.75, 2.75)	−1 (−4, 1.5)	−1 (−4.5, 3.5)	−1 (−3.75, 1)	−3 (−5, 0.25)	−3 (−7, 0)	−9 (−12.5, 0)	−1.5 (−3.25, 1)	−2.5 (−3, −0.5)	−3 (−4.5, −1.5)	
	*p*-value	0.727	0.003	0.344		0.678	0.824	0.549	1	0.03	0.004	0.003	0.118				
	HFMSE Median (IQR)	0 (−1.75, 1.75)	3 (1, 4)	3.5 (−2.25, 6)	6 (1, 8)	1 (−0.75, 3)	1.5 (−2.75, 3.75)	−0.5 (−3, 2)	2 (−1.5, 3.5)	−1 (−4, 0.75)	−3 (−6.25, 0)	−4.5 (−9.25, −0.75)	−6 (−10, −2)	−2 (−4, 1)	−2 (−2, −0.5)	−1.5 (−4.25, 1.25)	
	*p*-value	1	0.006	0.344		0.064	0.189	0.754	0.344	0.007	0.001	0.001	0.035				
3b	M	1	0	0	0	1	2	0	0	9	7	5	4	15	6	5	6
	N	1	0	0	0	1	1	0	0	3	3	3	2	5	3	3	1
	RHS Median (IQR)						2 (2, 2)			−3 (−8, 0)	1 (−4, 3.5)	3 (−2, 3)	0 (−5.25, 5.75)	−1 (−1.5, 0.5)	1 (−2.75, 4.75)	1 (−1, 5)	0.5 (−0.75, 2.5)
	*p*-value																
	HFMSE Median (IQR)						3 (3, 3)			1 (−2, 2)	1 (−1.5, 2)	1 (−1, 3)	1.5 (−1.25, 3.5)	0 (−1, 1)	0.5 (−2.25, 1.75)	0 (−1, 2)	−0.5 (−1, 3.75)
	*p*-value																

**Table 4 jcm-12-01920-t004:** Up to 2-year median change in the RHS and HFMSE cross-tabulated by WHO-derived functional type and age. *N*: number of participants, *M*: number of assessments, *IQR*: inter-quartile range.

Age		<5				5–7				8–13				14–18			
Functional Type		6 m	12 m	18 m	24 m	6 m	12 m	18 m	24 m	6 m	12 m	18 m	24 m	6 m	12 m	18 m	24 m
Non-Sitter	M	3	3	1	0	1	0	1	0	9	10	6	4	1	2	1	0
	N	3	2	1	0	1	0	1	0	6	8	4	4	1	2	1	0
	RHS Median (IQR)	0 (−0.5, 1.5)	5 (3.5, 5.5)							0 (0, 1)	−1 (−1, 0)	−0.5 (−1, 0)	−0.5 (−1, 0.25)		0 (0, 0)		
	*p*-value										0.289						
	HFMSE Median (IQR)	1 (0.5, 1.5)	7 (6, 10)							0 (0, 1)	−1 (−1, 0)	0 (0, 0)	0 (−0.25, 0.25)		0 (0, 0)		
	*p*-value										0.125						
Sitter	M	81	61	26	22	38	23	8	9	56	45	29	21	13	9	5	1
	N	33	27	17	12	20	15	7	7	28	22	14	13	7	7	4	1
	RHS Median (IQR)	0 (0, 2)	1 (0, 2)	2.5 (1, 5)	1.5 (1, 5.75)	0 (−1, 0.75)	0 (−1, 1.5)	−1 (−3.25, 0)	−1 (−2, 0)	−1 (−1.25, 0.25)	−1 (−3, 0)	−2 (−4, 0)	−2 (−5, 0)	0 (−1, 1)	−1 (−1, 0)	0 (0, 0)	
	*p*-value	0.014	*p* < 0.001	*p* < 0.001	*p* < 0.001	0.832	1		0.289	0.032	0.002	0.023	0.013	1	0.453		
	HFMSE Median (IQR)	0 (0, 3)	1 (0, 3)	3 (0, 6.5)	2.5 (1, 6)	0 (−1, 1)	−1 (−2, 1)	−0.5 (−3.25, 0.25)	−2 (−4, 1)	−1 (−3, 0)	−2 (−5, 0)	−4 (−7, −2)	−5 (−8, −1)	0 (−2, 0)	−1 (−2, 0)	−1 (−1, 0)	
	*p*-value	0.018	0.002	0.004	*p* < 0.001	0.69	0.664	0.687	0.508	*p* < 0.001	*p* < 0.001	*p* < 0.001	0.001	0.219	0.453		
Transitional Group	M	5	4	2	1	5	6	7	4	10	9	5	2	4	2	0	0
	N	2	2	1	1	2	3	3	2	5	7	5	2	3	2	0	0
	RHS Median (IQR)	1 (0, 2)	−1.5 (−2.25, −0.25)	−1.5 (−1.75, −1.25)		−1 (−3, 0)	−3 (−4.5, −2.25)	−4 (−6, −3)	−7 (−7.5, −6.25)	−2.5 (−3.75, −0.25)	−3 (−6, −2)	−7 (−7, −7)	−6 (−6.5, −5.5)	−0.5 (−2.75, 1)	−3.5 (−3.75, −3.25)		
	*p*-value										0.004						
	HFMSE Median (IQR)	1 (0, 1)	−1.5 (−3, 0.25)	−2.5 (−2.75, −2.25)		1 (−4, 3)	−3.5 (−5.5, −1.5)	−5 (−6, −3)	−4 (−5.5, −4)	−2.5 (−4.75, −0.25)	−5 (−9, −3)	−11 (−11, −9)	−8.5 (−10.25, −6.75)	−2 (−3.75, 0.25)	−1 (−1.5, −0.5)		
	*p*-value										0.004						
Walker	M	11	12	10	6	22	19	8	9	43	33	23	16	14	6	5	6
	N	7	9	9	5	10	13	5	8	17	14	15	13	4	3	3	1
	RHS Median (IQR)	0 (−2.5, 3)	5 (3.5, 8)	4 (−1.5, 5.75)	6 (3.75, 7.5)	1 (−3.5, 3)	1 (−3, 2.5)	0.5 (−1, 3.75)	−1 (−3, 4)	−1 (−3, 1)	0 (−5, 2)	−2 (−5.5, 1.5)	−7.5 (−11.25, 3.25)	−1 (−1.75, 0)	1 (−2.75, 4.75)	1 (−1, 5)	0.5 (−0.75, 2.5)
	*p*-value	1	0.006	0.344		0.503	0.815		1	0.256	0.711	0.263	0.454				
	HFMSE Median (IQR)	0 (−1.5, 1.5)	3 (1, 4.5)	3.5 (−2.25, 6)	5.5 (0, 8)	1 (−0.75, 2)	2 (−2.5, 3.5)	0.5 (−1, 5.25)	2 (0, 4)	0 (−2, 1.5)	−1 (−3, 2)	−1 (−5.5, 1)	−4 (−10, 1.25)	0 (−0.75, 1)	0.5 (−2.25, 1.75)	0 (−1, 2)	−0.5 (−1, 3.75)
	*p*-value	1	0.012	0.344		0.078	0.238		0.289	0.743	0.473	0.189	0.302				

**Table 5 jcm-12-01920-t005:** Up to 2-year median change in the RHS and HFMSE cross-tabulated by baseline RHS group and age. *N*: number of participants, *M*: number of assessments, *IQR*: inter-quartile range.

Age		<5				5–7				8–13				14–18			
RHS Baseline Total		6 m	12 m	18 m	24 m	6 m	12 m	18 m	24 m	6 m	12 m	18 m	24 m	6 m	12 m	18 m	24 m
Q1: 0–4	M	17	20	8	9	8	3	1	0	19	19	11	9	8	6	4	1
	N	9	10	6	3	3	1	1	0	11	11	6	8	5	5	3	1
	RHS Median (IQR)	1 (0, 2)	1 (0.75, 1)	1 (0.75, 3)	1 (1, 2)	0 (−1, 0)	0 (−0.5, 0)			0 (0, 0.5)	0 (−1, 0)	0 (0, 0.5)	0 (−1, 0)	0 (−1, 1)	0 (−0.75, 0)	0 (0, 0)	
	*p*-value	0.002	0.001	0.031						1	0.388						
	HFMSE Median (IQR)	1 (0, 2)	0.5 (0, 3)	1 (0, 3.5)	2 (0, 2)	0 (−0.25, 0.5)	1 (1, 1.5)			0 (0, 0)	−1 (−1, 0)	0 (−0.5, 0)	0 (−1, 0)	0 (0, 0)	0 (0, 0)	−0.5 (−1, 0)	
	*p*-value	0.065	0.092							1	0.012						
Q2: 5–9	M	39	25	10	5	15	7	2	3	16	12	8	5	1	2	0	0
	N	18	11	6	4	11	5	2	3	10	7	4	2	1	2	0	0
	RHS Median (IQR)	0 (0, 2)	2 (0, 5)	5.5 (1.5, 7.75)	8 (7, 9)	0 (−0.5, 1)	2 (0, 2)	1 (0.5, 1.5)	0 (−0.5, 1)	0 (−1, 0.25)	−1 (−2, −0.75)	−2 (−3.25, −0.5)	−2 (−3, −1)		0.5 (−0.25, 1.25)		
	*p*-value	0.076	0.004	0.004		0.754				0.549	0.065						
	HFMSE Median (IQR)	1 (0, 3)	2 (0, 8)	6 (1.5, 10.25)	9 (9, 11)	0 (−1, 0.5)	0 (−1.5, 2)	1 (1, 1)	1 (−0.5, 2)	−1 (−2.5, 0)	−2.5 (−4, −1)	−4.5 (−6, −4)	−6 (−8, −4)		0 (−1, 1)		
	*p*-value	0.043	0.004	0.002		0.754				0.065	0.012						
Q3: 10–18	M	27	18	9	7	14	12	6	5	20	16	12	11	1	1	2	0
	N	14	11	8	5	8	10	5	3	11	9	8	7	1	1	2	0
	RHS Median (IQR)	0 (−1, 1)	0 (−1, 1.75)	2 (−1, 3)	1 (0.5, 2)	0 (−0.75, 0)	0 (−1, 0.25)	−2.5 (−3.75, −0.5)	−1 (−3, −1)	−1 (−2.25, 1)	−2.5 (−5, −0.75)	−3 (−4.75, −0.75)	−5 (−7, −1.5)			−3 (−4.5, −1.5)	
	*p*-value	0.824	1	0.508		1	1			0.332	0.035	0.021	0.065				
	HFMSE Median (IQR)	0 (−1.5, 1)	0.5 (−0.75, 3)	1 (−3, 3)	3 (1, 5)	0 (−2, 1)	−1 (−2.5, 0.25)	−2 (−3.75, −0.25)	−3 (−4, −2)	−1.5 (−3, 0.5)	−3 (−7.25, −1.75)	−4.5 (−9.5, −2.75)	−6 (−11.5, −3.5)			−1.5 (−4.25, 1.25)	
	*p*-value	0.824	0.424	0.727		1	0.227			0.064	0.007	0.012	0.065				
Q4: 19–42	M	12	14	8	6	10	12	8	6	30	25	15	9	10	6	1	0
	N	7	8	6	4	5	8	4	4	16	16	10	7	4	4	1	0
	RHS Median (IQR)	0.5 (−0.25, 2)	4 (0.25, 7.5)	3.5 (−1.25, 6)	7 (6, 9.5)	−1.5 (−3.75, 0.75)	−2.5 (−5.25, −1.25)	−4 (−6, −2)	−5.5 (−7, −2.5)	−2 (−3.75, −0.25)	−4 (−6, −1)	−6 (−10.5, −4.5)	−9 (−10, −5)	0.5 (−2.5, 1)	−1.5 (−3, 3)		
	*p*-value	0.508	0.092	0.727			0.146			0.004	0.001	0.001	0.18				
	HFMSE Median (IQR)	0.5 (0, 1.25)	1.5 (−0.5, 3.75)	3 (−2.25, 6)	4.5 (0, 7.5)	1 (−3, 1.75)	−3.5 (−4.25, 1.25)	−4 (−5.5, −2.25)	−4 (−4.75, −4)	−2 (−4, 0)	−5 (−9, −1)	−9 (−11, −5.5)	−6 (−10, −2)	−0.5 (−3.5, 0.75)	−1 (−2, 0.75)		
	*p*-value	0.289	0.18	0.727			0.388			0.019	*p* < 0.001	0.002	0.07				
Q5: 43–69	M	5	3	4	2	19	14	7	8	33	25	17	9	12	4	4	6
	N	4	3	4	2	9	9	4	7	12	10	11	9	4	2	2	1
	RHS Median (IQR)	−1 (−4, 3)	4 (3, 10.5)	−1 (−6.5, 4.25)	−6 (−10.5, −1.5)	1 (−3, 2.5)	1 (−3.5, 2)	0 (−1, 2)	−1.5 (−3.75, 4.5)	0 (−2, 2)	0 (−4, 3)	0 (−3, 2)	−5 (−15, 3)	−1 (−2, −1)	−2.5 (−3.5, −0.25)	0 (−1.75, 2)	0.5 (−0.75, 2.5)
	*p*-value					0.629	0.581		0.727	0.711	1	0.791	1				
	HFMSE Median (IQR)	−1 (−2, −1)	1 (0.5, 5)	−1 (−6.75, 4.75)	0 (−4, 4)	1 (−2, 2.5)	2.5 (−1.5, 3.75)	0 (−1, 3.5)	2 (−0.75, 5)	0 (−2, 2)	0 (−2, 3)	−1 (−5, 1)	−3 (−9, 2)	0 (−2.25, 0.25)	−1.5 (−3.5, 1.25)	−0.5 (−1.5, 1)	−0.5 (−1, 3.75)
	*p*-value					0.238	0.267		0.453	0.851	1	0.454	1				

**Table 6 jcm-12-01920-t006:** Breakdown of RHS and HFMSE total scores in assessments where at least one was 0.

		HFMSE Total Score	
		0	1
RHS Total	0	6	2
	1	7	0
	2	6	0

## Data Availability

The data presented in this study are available on request from the corresponding author after steering committee approval. The data are not publicly available due to ethics.

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
