# Peer review of "2-Year Change in Revised Hammersmith Scale Scores in a Large Cohort of Untreated Paediatric Type 2 and 3 SMA Participants"

_jcm, 2023, doi:10.3390/jcm12051920_

Round 1

Reviewer 1 Report

This very interesting paper presents a 2-year change in RHS assessments in a pediatric not-treated cohort of 177 SMA 2 and 3 patients. The results concerning natural history are in general agreement with the previous studies of Corrati et al 2020 and Mercuri et al 2016,  with possible improvement in the first 5 years of life and the highest decline between 8-13. The authors consider a new transitional group of patients (crawlers,

The article is well written however not easy to read (due to numbers and statistics).  

The authors suggest the RHS assessment even for the weakest patients.  But participants scoring in the lowest quartile had no detectable change in RHS, and patients with initial results 0 could not decline at all. Isn’t it more reasonable to use CHOP-Intend in this group?

Minors:  

Table 1. The column with the range of age should be added (there is only median age).  

Table 6. The content and description of table 6 should be more clear, it is not understandable 

Line 454-460-  sentence is repeated 

Table legends are not given (abbreviations N, M, IQR) 

Author Response

This very interesting paper presents a 2-year change in RHS assessments in a pediatric not-treated cohort of 177 SMA 2 and 3 patients. The results concerning natural history are in general agreement with the previous studies of Corrati et al 2020 and Mercuri et al 2016, with possible improvement in the first 5 years of life and the highest decline between 8-13. The authors consider a new transitional group of patients (crawlers,

The article is well written however not easy to read (due to numbers and statistics).  

The authors suggest the RHS assessment even for the weakest patients.  But participants scoring in the lowest quartile had no detectable change in RHS, and patients with initial results 0 could not decline at all. Isn’t it more reasonable to use CHOP-Intend in this group?

Thank you for raising this interesting question of the CHOP-Intend – we have added context for when the RHS should be used on line 378-381. The administration of the full CHOP-Intend score becomes difficult as the child gets heavier because of items 15 and 16 which are performed in ventral suspension. This results in partial assessments of the CHOP-Intend in older children, and therefore we would not recommend the leading to incomplete CHOP-intend assessments. We believe that the RULM would be more appropriate for these weakest children, as we have seen that there is more variability in scores, but that the RHS is more appropriate than the HFMSE thanks to the reduced floor effect.

Minors:  

Table 1. The column with the range of age should be added (there is only median age).  

Thank you for identifying this omission, we have included the column in Table 1.

Table 6. The content and description of table 6 should be clearer, it is not understandable 

Thank you for identifying this, we have made this clearer by expanding the description of the table. We are using this table to compare how patients who score a 0 on the HFMSE performed on the RHS, and how patients who scored a 0 on the RHS performed on the HFMSE.

Line 454-460- sentence is repeated 

Thank you for identifying this, we have removed the extra line.

Table legends are not given (abbreviations N, M, IQR) 

Thank you for identifying this, we have added the explanation for the abbreviations to each of the tables

Reviewer 2 Report

This well-written important study includes a multinational comprehensive natural history dataset to contextualise disease progression in different subgroups of the SMA2 and SMA3 cohorts. This study addressed two important questions: 1). 2 years change of RHS scores to study the natural history of SMA 2 & 3, including a newly defined transition group; 2). Comparing RHS to HFMSE

The following significant findings have been stated. For the sitters broken down by age groups, there was a significant change in RHS median scores of 1.5 at 24 months for the under 5, while there was a non-significant trend towards the decline of -1 and -2 at 24 months in the 5-7 and 8-13 age groups, respectively. When considered the newly defined transitional group, this group displayed the greatest change over time and displayed a relatively homogeneous decline beyond the age of 5.

This study also showed that RHS is more effective at differentiating non-sitter participants and has a reduced floor effect compared to the HFMSE. In addition, the study findings suggested that the RULM should be used with the RHS for the weaker patients, especially those scoring less than 20 on the RHS. Moreover, participants with the same RHS total can be differentiated by their timed test items. While the RHS over time is as sensitive as the HFMSE, both can detect changes in certain type, age and motor function subgroups. 

With much information presented in this manuscript, the focus on comparing RHS and HFMSE dominates the presentation. Therefore, it will be helpful to guide the readers in the abstract with some rearrangement of the current version to highlight more the natural history findings for SMA2/3a and the transition group in the change of the RHS scores before and after the age of 5. It will also be helpful to the reader if the authors comment on whether the Revised HS (RHS) is preferred when monitoring the SMA 2/3 patients for both clinical and research purposes. 

What are the changes in the RHS scores for the SMA 3b group? Please also consider having a brief elaboration on this in the manuscript. 

Author Response

This well-written important study includes a multinational comprehensive natural history dataset to contextualise disease progression in different subgroups of the SMA2 and SMA3 cohorts. This study addressed two important questions: 1). 2 years change of RHS scores to study the natural history of SMA 2 & 3, including a newly defined transition group; 2). Comparing RHS to HFMSE

The following significant findings have been stated. For the sitters broken down by age groups, there was a significant change in RHS median scores of 1.5 at 24 months for the under 5, while there was a non-significant trend towards the decline of -1 and -2 at 24 months in the 5-7 and 8-13 age groups, respectively. When considered the newly defined transitional group, this group displayed the greatest change over time and displayed a relatively homogeneous decline beyond the age of 5.

This study also showed that RHS is more effective at differentiating non-sitter participants and has a reduced floor effect compared to the HFMSE. In addition, the study findings suggested that the RULM should be used with the RHS for the weaker patients, especially those scoring less than 20 on the RHS. Moreover, participants with the same RHS total can be differentiated by their timed test items. While the RHS over time is as sensitive as the HFMSE, both can detect changes in certain type, age and motor function subgroups. 

With much information presented in this manuscript, the focus on comparing RHS and HFMSE dominates the presentation. Therefore, it will be helpful to guide the readers in the abstract with some rearrangement of the current version to highlight more the natural history findings for SMA2/3a and the transition group in the change of the RHS scores before and after the age of 5.

Thank you for this helpful comment, we have added a comment into the abstract [Line 43-45], which describes from a high-level the observed changes in the RHS, with relation to motor function and age group.

It will also be helpful to the reader if the authors comment on whether the Revised HS (RHS) is preferred when monitoring the SMA 2/3 patients for both clinical and research purposes. 

Thank you for this guidance, we have added a comment on this to the end of the conclusion [Line 481-483].

What are the changes in the RHS scores for the SMA 3b group? Please also consider having a brief elaboration on this in the manuscript. 

Thank you for this comment- we have addressed this in two places – line 422-424 and line 447-449. There is stability in the SMA 3b group, although small sample sizes made it not possible to see trends with age. In this group, it may be possible to identify a change in up-to 2-years based on change in timed tests, but this was not possible using the current data.